# One-Step Construction of Multi-Walled CNTs Loaded with Alpha-Fe_2_O_3_ Nanoparticles for Efficient Photocatalytic Properties

**DOI:** 10.3390/ma14112820

**Published:** 2021-05-25

**Authors:** Jianle Xu, Qiang Wen, Xiao Zhang, Yinhui Li, Zeyue Cui, Pengwei Li, Chunxu Pan

**Affiliations:** 1Center of Nano Energy and Devices, College of Information and Computer, Taiyuan University of Technology, Taiyuan 030000, China; xujianle@whu.edu.cn (J.X.); wenqiang@tyut.edu.cn (Q.W.); liyinhui@tyut.edu.cn (Y.L.); cuizheyue@tyut.edu.cn (Z.C.); 2School of Physics and Technology, and MOE Key Laboratory of Artificial Micro, and Nano-Structures, Wuhan University, Wuhan 430072, China; cxpan@whu.edu.cn; 3College of Environmental Science and Engineering, Taiyuan University of Technology, Taiyuan 030000, China; zhangxiao02@tyut.edu.cn

**Keywords:** CNTs, alpha-Fe_2_O_3_, synergistic effect, heterostructure, photocatalytic properties

## Abstract

The aggregation and the rapid restructuring of the photoinduced electron−hole pairs restructuring in the process of photoelectric response remains a great challenge. In this study, a kind of Multi-walled carbon nanotubes loaded Alpha-Fe_2_O_3_ (CNTs/α-Fe_2_O_3_) heterostructure composite is successfully prepared via the one-step method. Due to the synergistic effect in the as-prepared CNTs/α-Fe_2_O_3_, the defect sites and oxygen-containing functional groups of CNTs can dramatically improve the interface charge separation efficiency and prevent the aggregation of α-Fe_2_O_3_. The improved photocurrent and enhanced hole–electron separation rate in the CNTs/α-Fe_2_O_3_ is obtained, and the narrower band gap is measured to be 2.8 ev with intensive visible-light absorption performance. Thus, the CNTs/α-Fe_2_O_3_ composite serves as an excellent visible light photocatalyst and exhibits an outstanding photocatalytic activity for the cationic dye degradation of rhodamine B (RhB). This research supplies a fresh application area forα-Fe_2_O_3_ photocatalyst and initiates a new approach for design of high efficiency photocatalytic materials.

## 1. Introduction

Currently, environmental pollution has become a potential threat to the sustainable development of human ecology. Utilization of solar energy and photocatalyst degrades organic pollutants and produces hydrogen by water splitting, showing great potential in energy conversion and environmental governance [1,2,3]. It is important that one of the greatest challenges under the circumstances of the photo-catalysis is how to optimize the efficiency of photo-catalysts to reduce the recombination of photogenerated charge carriers and enhance the absorption intensity and range of excitation light, which will depend on the types and performances (e.g., crystalline structure, width of the band gap, specific surface and so on) of the photocatalyst. Up to now, many efforts have been dedicated to searching for high photocatalytic activity materials such as n-type semiconductor materials TiO_2_ [4], ZnO [5], SnO_2_ [6], Fe_2_O_3_ [7], etc. A great deal of work has been done on the preparation of pure α-Fe_2_O_3_ for photocatalysis, such as the use of ultrasonic-assisted method [8], etc. Compared with other preparation methods, the one-step synthesis method has significant advantages, as the experimental operation is simple and feasible, and the synthesized nanoparticles have small particle size and good dispersion.

Nano iron oxides are widely used in heterogeneous catalyst, and alpha-iron oxide (α-Fe_2_O_3_) has been considered as one of the significant members in the energy conversion filed due to its favorable band gap, stability, and abundance. By its narrow band gap and high stability, it absorbs visible light and other sources. However, as α-Fe_2_O_3_ has denied electron-hole pair recombination [9,10], the rapid reorganization recombination of the photoinduced electron−hole pairs reduces photocatalytic performance expressly, and the photocatalytic reaction is slightly affected [11,12].

It is well known that making full use of photogenerated carriers is pivotal to improving the activity of photocatalyst based on semiconductor photocatalytic mechanism. Recently, several strategies have been studied extensively, showing that the recombination of the photo-generated electrons and holes is able to be restrained by constructing of heterojunction [13,14], development of Z-scheme heterojunction [15,16] and coupling of support materials [17,18,19]. Combining of carbon-based materials (e.g., activated carbon, graphene, carbon quantum dots, CNTs) for catalyst has attracted much attention as new-fire strategy to transfer of photoinduced charge carriers and increase lifetime during the photocatalytic activity. There have been many reports on the synthesis of CNTs/α-Fe_2_O_3_ nanocomposite [20]. According to a recent study, CNTs have increased the photocatalytic performance because of excellent electron transport capability and enhanced separation rate of charge carriers [21,22,23,24,25]. Moreover, the oxygen-containing functional groups and defect sites on the outside of carbon materials can also effectively inhibit the agglomeration of semiconductor nanoparticles (NPs) [26,27]. To the best of our knowledge, heterostructure hybrids designed based on Multi-walled CNTs and α-Fe_2_O_3_ for photocatalytic have rarely been reported. Moreover, limited progress has been made towards a basic understanding of photocatalytic mechanism in α-Fe_2_O_3_ heterostructure materials. Thus, designing α-Fe_2_O_3_@carbon-based materials (e.g., activated carbon, graphene, carbon quantum dots, and CNTs) will be an effective pathway to improving photochemical and catalytic properties [21].

Herein, we reported a novel and simple strategy to solve the aforementioned defect of the α-Fe_2_O_3_ in photocatalytic filed. The CNTs/α-Fe_2_O_3_ composite has been synthesized by a one-step hydrothermal method successfully. This work demonstrates that CNTs/α-Fe_2_O_3_ composite exhibits greatly improved photocatalytic activity on the degradation of RhB attributed to improved photocurrent and enhanced hole-electron separation rate, when compared to single α-Fe_2_O_3_. Meanwhile, this facile method for novel heterojunction photocatalysts may be widely used in the field of water treatment.

## 2. Materials and Methods

### 2.1. Materials

Multi-walled CNTs, with size 0.5–2 μm in length and 30–50 nm in diameter, were purchased from the Chinese Academy of Science, Chengdu Organic Chemistry Co., Ltd., Chengdu, China, Ferric chloride hexahydrate (FeCl_3_·6H_2_O, 99%), phenol (C_6_H_6_O, 99.5%), sodium acetate (CH3COONa, 99%), rhodamine B (C_28_H_31_ClN_2_O_3_, RhB), methyl orange (C_14_H_14_N_3_SO_3_Na, MO), congo red (C_32_H_22_N_6_Na_2_O_6_S_2_, CR), and methylene blue (C_16_H_18_CIN_3_S, MB) were from Sinopharm Chemical Reagent Co., Ltd. (Taiyuan, China). No further purification was done before the use of all the reagents and chemicals.

### 2.2. Synthesis of α-Fe_2_O_3_ Nanoparticles

The α-Fe_2_O_3_ NPs were prepared using a method reported by our previous work [9]. 0.02 M FeCl_3_·6H_2_O and 0.1 M CH_3_COONa was dispersed into 60 mL deionized water. Then, the suspension was transferred and sealed in a stainless-steel autoclave with 250 °C for 5 h. Lastly, the obtained sample was dried in vacuum oven at 60 °C.

### 2.3. Synthesis of CNTs/α-Fe_2_O_3_ Composite Nanomaterials

The CNTs/α-Fe_2_O_3_ composite was prepared via the one-step hydrothermal process. Firstly, 0.324 g of FeCl_3_·6H_2_O and 0.492 g CH_3_COONa was dispersed into 30 mL deionized water. After that, CNTs with different content (2.5%, 5%, 7.5%) were added into the above solution. After ultrasonic process for 30 min, the mixed sample was transferred into a stainless-steel autoclave, which was kept at 250 °C for 5 h. After the reaction, the product was harvested through several rinse−precipitation cycles with deionized water and ethanol and dried at 60 °C over-night. The CNTs/α-Fe_2_O_3_ composites finally were obtained. The obtained products (CNTs-loaded with α-Fe_2_O_3_NPs) corresponding to proportion of CNTs addition (2.5%, 5%, 7.5%) were denoted as CNTs/α-Fe_2_O_3_: S_1_-S_3_, respectively.

### 2.4. Characterization

The structure of samples was measured by X-ray diffraction (XRD) patterns which were provided by a powder diffractometer (Bruker AXS, Karlsruhe, Germany). The structure and morphology were measured by scanning electron microscopy (SEM, JSM-7001F, JEOL, Tokyo, Japan) and transmission electron microscopy (TEM, 2100F, JEOL, Tokyo, Japan). Energy-dispersive X-ray analysis (EDX, INCA Penta FETx3) was provided with the SEM. Raman spectra were measured by the spectrometer (LabRAM HR 800UV, HPRIBA JOBIN YVON, Paris, France). UV–VIS spectra were measured using the spectrophotometer (3100 UV–VIS-NIR, Shimadzu, Kyoto, Japan).

### 2.5. Photoelectrochemical Performance of α-Fe_2_O_3_ and CNTs/α-Fe_2_O_3_

The photoelectrochemical (PEC) is measured by an electrochemical workstation (IM6, Zahner, Karlsruhe, Germany) with a 500 W Xe light furnished with a cut-off filter (l > 420 nm) used for the source of visible light. A saturated Ag/AgCl and Pt wire was used as a counter and a reference electrode, indium-tin oxide (ITO) glass, as working electrode. Then, 1 mg CNTs/α-Fe_2_O_3_, 5 wt.% Nafion (10 mL), ethanol (0.2 mL), and water (0.2 mL) were mixed to form a suspension. After ultrasonication for 30 min, the suspension was coated on the surface of ITO glasses (2.0 cm × 2.0 cm) and dried at 60 °C for several hours to prepare the photoelectrodes. In the PEC measurements, Na_2_SO_4_ (0.5 M) was adopted as the electrolyte. the photocurrent properties of the photoanode were evaluated by chronoamperometric I-t curves measured with interval time of 60 s.

For the test of photocatalytic performance, photocatalyst (5 mg) was dissolved into Phenol, RhB MB, MO (50 mL), and CR (0.01 mM), respectively, in order to produce the suspension by magnetically stirring continually in a dark environment for 1 h to reach adsorption–desorption equilibrium. After that, the 0.3 mL of hydrogen peroxide solution (H_2_O_2_, 30 wt.%) was added into above suspension with a refrigerating plant under magnetic stirring, and the vertical distance between the lamps was 50 cm (the energy directly radiated was about 95 mW/mL). During the experiment, every 30 min, 3 mL of solution was extracted from the suspension, and the change of the maximum absorbance was measured by spectrophotometer, while the concentration of dye in the filtrate at different intervals was measured.

## 3. Results and Discussion

Figure 1 shows that structure and composition characterization of CNTs/α-Fe_2_O_3_ composite highly regular α-Fe_2_O_3_ NPs grow on the CNTs. The morphology of microstructures of the CNTs and CNTs/α-Fe_2_O_3_ were investigated by SEM, as shown in Figure 1a,b. One could find that a large number of α-Fe_2_O_3_ NPs adhere to the surface of CNTs. As shown in Figure 1c,d, the detailed microstructures of the CNTs/α-Fe_2_O_3_ composite was conducted by low-magnification TEM and high-resolution TEM. Significantly, the α-Fe_2_O_3_ NPs on the surface of CNTs can be observed clearly. Moreover, one can find that α-Fe_2_O_3_ with an average diameter of 20–30 nm was uniformly deposited on the outside of CNTs after a hydrothermal process. The average diameter of the CNTs was 30–50 nm, and the length was about 1–2 μm. As shown in Figure 1d, the clear HRTEM image observations reveal that the CNTs/α-Fe_2_O_3_ heterostructure nanocrystals are composed of CNTs and α-Fe_2_O_3_ NPs. Furthermore, HRTEM image of the typical hybrid nanostructure exhibits a single-crystalline structure with the spacing of 0.27 nm, corresponding to the {110} planes of α-Fe_2_O_3_ [28], and the interlayer spacing of graphite in Multi-walled CNTs is 0.34 nm [29]. No other contaminations are observed, possibly indicating that their interface between the CNTs and α-Fe_2_O_3_ NPs is “clean”.

Figure 2a shows XRD spectra of the single CNTs, α-Fe_2_O_3_ and CNTs/α-α-Fe_2_O_3_-Sx composite. All the diffraction peaks are indexed by the rhombohedral α-Fe_2_O_3_ phase (JCPDS card No. 33-0664) without any peaks of other phases, which indicates the enhanced purity of as-prepared α-Fe_2_O_3_. The XRD pattern of the CNTs/α-Fe_2_O_3_ composite has a similar diffraction to α-Fe_2_O_3_. However, unobserved CNTs signal in the XRD pattern indicates the amount of CNTs in the sample was very low and the loading content of CNTs didn’t affect the crystal structure of α-Fe_2_O_3_ [30]. More composition information can be confirmed by the Raman spectra, the Raman spectra of α-Fe_2_O_3_ [31], CNTs and CNTs/α-Fe_2_O_3_-Sx in a range of 800–1800 cm^−1^ as shown in Figure 2b. Raman spectrum of CNTs/α-α-Fe_2_O_3_-Sx with two peaks at 1330 and 1585 cm^−1^ assigned to D and G bands confirms the presence of CNTs in the nanocomposite [32]. D and G bands arise from the defects and disorder in carbon materials and stretching mode of C-C bonds in the graphite crystallites, respectively [33,34]. For pure CNTs, the *I_D_/I_G_* value is 1.23. In the presence of α-Fe_2_O_3_, the graphitization degree of CNTs was improved, and the *I_D_/I_G_* value is about 1.1. Based on the above analysis, the assumption that the amount of CNTs in the sample was very low proved to be correct.

The photoelectrochemical (PEC) performance of α-Fe_2_O_3_ with different loading amounts of CNTs can account for the photocatalytic activity (Figure 3a). With the periodic visible light irradiation at 60 s intervals in a conventional three-electrode system, the transient photocurrent values of the work electrodes were measured. The photocurrent density of CNTs/α-Fe_2_O_3_-S_2_ reached 0.34 mA/cm^2^ and was about 2.8 times as high as that of pure α-Fe_2_O_3_. This superior performance could be attributed to the introduction of CNTs, leading to the formation of synergistic effect. This synergistic effect makes it easier for CNTs to capture charge and improves photoelectron generation [35,36]. However, the photocurrent density of the composite decreases again, with a high CNTs loading (beyond 5wt% of CNTs/α-Fe_2_O_3_). This inhibitory effect may be associated with the balance of synergistic between CNTs and α-Fe_2_O_3_ [19,37]. The optical performance and photo absorption capacity of as-prepared nanocomposite were evaluated by the UV–S diffuse reflectance spectra. The visible light source is a 500W Xe (the wavelength distributes from 400 to 800 nm) light furnished (λ > 420 nm). In addition, the measured power of visible light irradiation on the sample is about 95 mW/mL. Here, pure α-Fe_2_O_3_ and CNTs/α-Fe_2_O_3_-S_2_ are assumed to be a direct band gap semiconductor [21]. There is a close relationship between the band gap energies (Eg) of absorbed wavelength range and semiconductors. The value of Eg are calculated by Kubelka–Munk. Figure 3b clearly displays the visual absorption peak of the single α-Fe_2_O_3_ at corresponding to band gap energy of 2.37 eV (Figure 3c). However, the absorption peak of typical CNTs/α-Fe_2_O_3_-S_2_ has about 23 nm red shift, which is 457 nm. The band gap energy of typical CNTs/α-Fe_2_O_3_-S_2_ is 2.08 eV which is smaller than that of pure α-Fe_2_O_3_. The course of band gap narrowing in CNTs/α-Fe_2_O_3_ is the surface defects formed of CNTs [38,39]. Obviously, the introduction of CNTs directly exhibits positive and improved impact on the photocatalytic properties and solar energy efficient utilization.

The photocatalytic activities of CNTs/α-Fe_2_O_3_ were measured by the degradation of RhB under visible light irradiation in aqueous solution. The dye concentration was analyzed by measuring the absorption intensity of RhB at 554 nm. With the extension of irradiation time, the coloration and absorption peak intensity of RhB solution decreased gradually, which means the concentration of RhB is continuously reduced. Figure 4a,b shows the variation of the RhB relative concentrations (C_t_/C_0_, and −Ln (C_t_/C_0_)) as a function of irradiation time, where C_t_ is the RhB concentration of the irradiation time and C_0_ is the initial concentration. Here, the main purpose of adding H_2_O_2_ is to make the whole reaction system in the Fenton system, Fe_2_O_3_ and H_2_O_2_, through Fenton reaction to generate strong oxidizing groups such as OH. In the presence of these strong oxidizing groups, the dye can be degraded. After irradiation for 5 h, only the photodegradation of RhB with H_2_O_2_ shows that the self-degradation can almost inappreciable, and 53%, 70.1%, and 49% RhB were degraded in the presence of CNTs/α-Fe_2_O_3_-S_1_, CNTs/α-Fe_2_O_3_-S_2_, CNTs/α-Fe_2_O_3_-S_3_, while there were degradation degrees of 46% and 9.1% by using the pure α-Fe_2_O_3_ and H_2_O_2_. Figure 4b indicates that the degradation rate constant of CNTs/α-Fe_2_O_3_-S_2_ reaches the maximum of 0.219/h. Obviously, when CNTs were loaded into α-Fe_2_O_3_, the photocatalytic of the prepared CNTs/α-Fe_2_O_3_-Sx samples had a prominent enhancement, especially with the loaded weight ratio of 5% (CNTs/α-Fe_2_O_3_-S_2_). This could be ascribed to the loading of CNTs prolonging the recombination time of electron-hole pairs [39]. For CNTs/α-Fe_2_O_3_, carriers can transfer from α-Fe_2_O_3_ to CNTs through the interface easily due to the defects of CNTs [25,40]. In cycle test experiments, samples were collected by centrifugation and washed before each test to ensure the purity. The RhB degradation ratio remained near 70% after 5 cycles (Figure 4c), indicating its excellent stability and photoactivity.

The photocatalytic activities of samples can be further evaluated by the degradation of dyes phenol, MB, MO, and CR. Figure 5 plots the comparison of photodegradation of different kinds of dyes. The degradation rate of CNTs/α-Fe_2_O_3_-S_2_ to cationic dye RhB, MB, neutral dye phenol (in the middle of the bar), and anionic dyes MO and CR were 70.1%, 60.2%, 41.2%, 31.8%, and 15.7%, respectively, which illustrates CNTs/α-Fe_2_O_3_ has a prominent photocatalytic performance in the cationic dyes [10].

In Figure 6, the mechanism of enhancing photocatalytic performance by CNTs loaded α-Fe_2_O_3_ was proposed based on the mentioned above analysis. Under the irradiation of sunlight, electrons from the valence band (VB) of the semiconductor are motivated into its conduction band (CB), meanwhile creating electron hole in the VB [10]. First, photoexcited electrons and the holes are rapidly recombined; then, only a few of the electrons and holes are involved in the reaction, which can be used to degrade pollutants. Secondly, when CNTs were loaded into α-Fe_2_O_3_, the free electrons can transfer from α-Fe_2_O_3_ to CNTs through the interface between them more easily due to the detects of CNTs, leading to a higher hole-electron separation rate [41,42,43]. Then, more carriers can be transferred to the active site to participate in the photocatalytic process, greatly reducing the electron-hole recombination process in α-Fe_2_O_3_. In conclusion, the main reaction process can be expressed as follows:CNTs/α-Fe_2_O_3_ + hv→α-Fe_2_O_3_(h^+^) + CNTs(e^−^)(1)
CNTs(e^−^) +O_2_ → CNTs + O^2−^(2)
α-Fe_2_O_3_+ H_2_O/OH-→α-Fe_2_O_3_ +OH(3)
OH/O^2−^ + pollutants → degradation products(4)

## 4. Conclusions

In the present research, CNTs/α-Fe_2_O_3_ nanocomposite was synthesized via a simple one-step hydrothermal method. The as-prepared CNTs/α-Fe_2_O_3_ composites possessed a heterojunction structure, where the α-Fe_2_O_3_ NPs were grown in the surface of Multi-walled carbon nanotubes. Because of the special structure of the composites, the high times photocurrent output value, and under visible light irritation, the narrower band gap (2.08 eV) and improved photocatalytic properties for the degradation of RhB dye, the obtained CNTs/α-Fe_2_O_3_ exhibited excellent photocatalytic performance. Thanks to the introduction of CNTs, the separation of photogenerated carriers is promoted. Taking advantage of this feature, introducing a certain quality ratio of CNTs into α-Fe_2_O_3_ can highly perfect the photocatalytic properties. The results of our experiments indicate that CNTs/α-Fe_2_O_3_ composites may be a promising photocatalyst to reduce the cost of the water treatment process.

## Figures and Tables

**Figure 1 materials-14-02820-f001:**
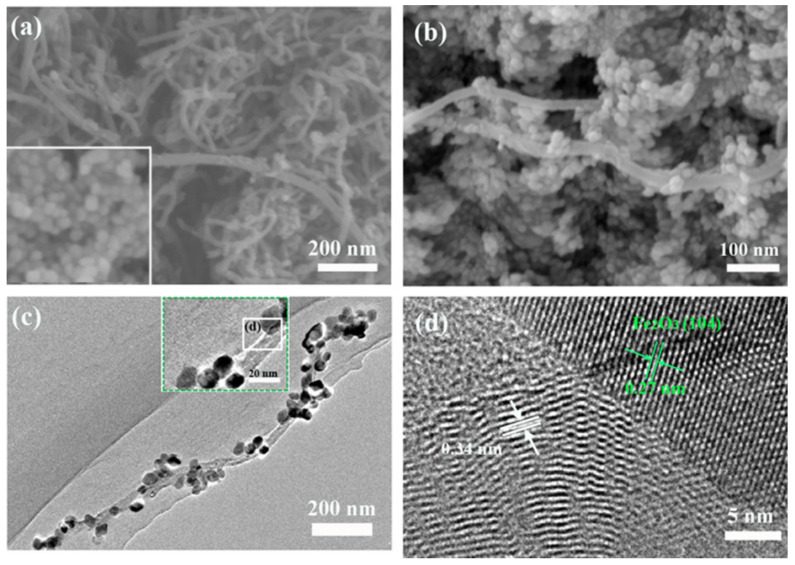
(**a**) SEM image of CNTs and pure α-Fe_2_O_3_ (inset); (**b**) CNTs/α-Fe_2_O_3_ NPs; (**c**) TEM image of CNTs/α-Fe_2_O_3_ nanocomposite (inset); (**d**) HRTEM image of CNTs/α-Fe_2_O_3_.

**Figure 2 materials-14-02820-f002:**
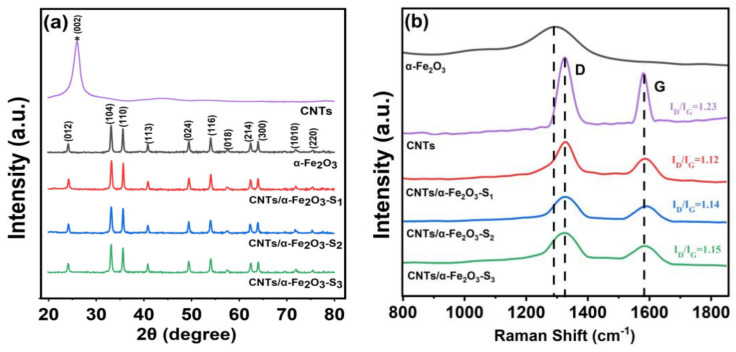
(**a**,**b**) XRD pattern and Raman spectra of CNTs, α-Fe_2_O_3_, and CNTs/α-Fe_2_O_3_-Sx composite S1, S2, and S3 are α-Fe_2_O_3_ with different mass ratio of CNTs loaded (2.5%, 5% and 7.5%).

**Figure 3 materials-14-02820-f003:**
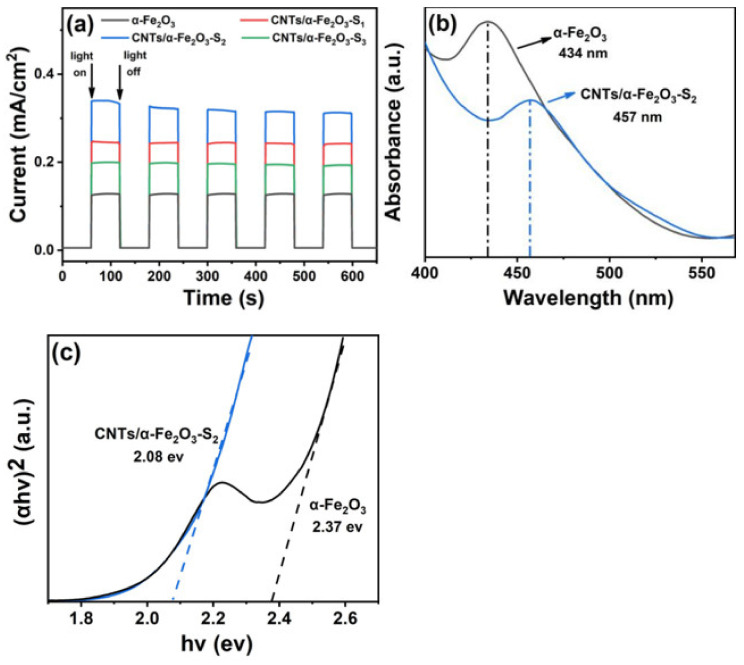
(**a**) Photocurrent density of α-Fe_2_O_3_ and CNTs/α-Fe_2_O_3_-S_x_; (**b**) UV–VIS absorption spectra for α-Fe_2_O_3_ and CNTs/α-Fe_2_O_3_-S_2_; (**c**) linear fits (dashed lines) of the (αhv) 2 hv curves.

**Figure 4 materials-14-02820-f004:**
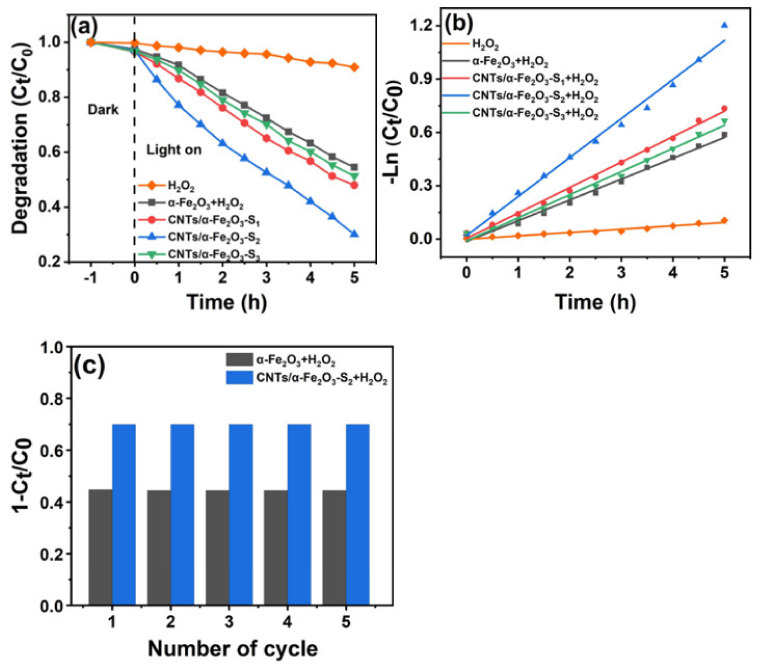
(**a**) Photocatalytic degradation of RhB with different samples; (**b**) kinetic curves of the RhB degradation; (**c**) cycle degradation of RhB.

**Figure 5 materials-14-02820-f005:**
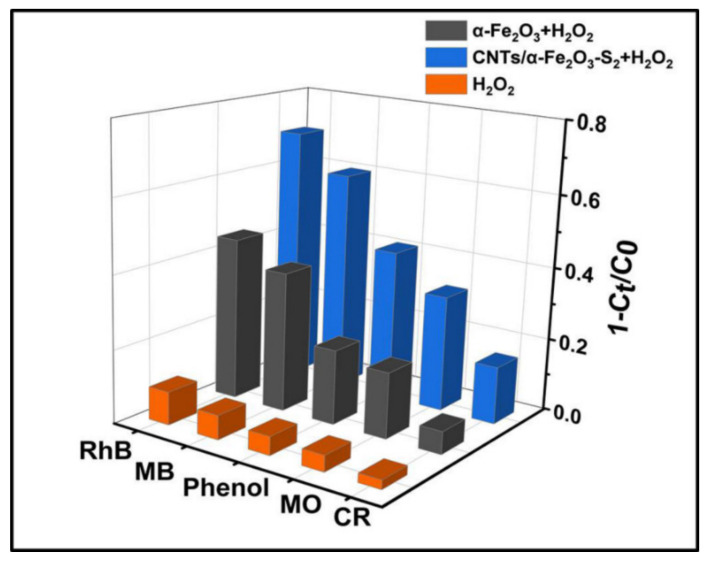
Photocatalytic degradation of several different pollutants with prepared photocatalysts.

**Figure 6 materials-14-02820-f006:**
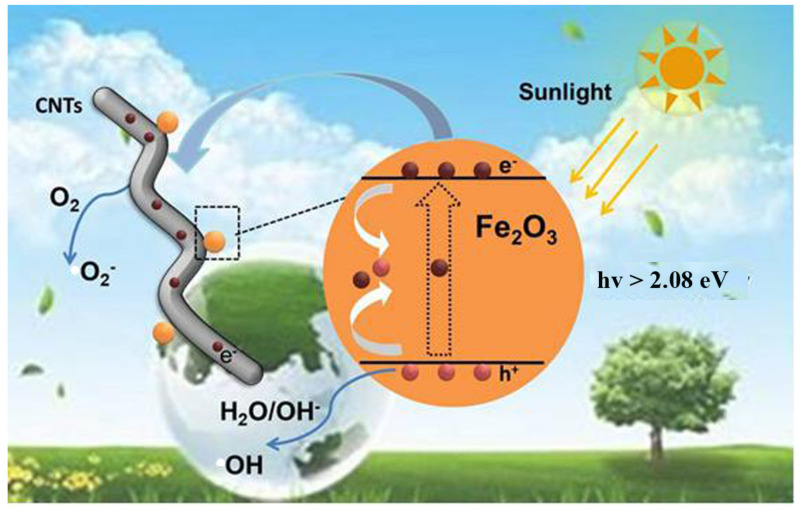
Schematic illustration of proposed photocatalytic mechanism for CNTs/α-Fe_2_O_3_.

## Data Availability

Data is contained within the article.

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
