# Peer review of "One-Step Construction of Multi-Walled CNTs Loaded with Alpha-Fe2O3 Nanoparticles for Efficient Photocatalytic Properties"

_materials, 2021, doi:10.3390/ma14112820_

Round 1

Reviewer 1 Report

In this work, the authors study the “One-step Construction of Multi-walled CNTs Loaded with Alpha-Fe2O3 Nanoparticles for Efficient Photocatalytic Properties”. They describe the method for the synthesis of Multi-walled CNTs Loaded with Alpha-Fe2O3 Nanoparticles and their characterization by SEM, TEM, UV-Vis, XRD. Photoelectrochemical and photocatalytic performance were tested. Results showed that photoelectrochemical and photocatalytic efficiency were influenced by weight percent CNTs.

The work can be accepted after minor revisions.

List of comments:

  1. In the abstract reference is made to the photovoltaic efficiency of the material, however in the text some photocurrent measurements are marginally shown. No recall is present in the conclusions. This reference should be removed from the abstract.
  2. Line 80. Insert some information on the method of synthesis of NPs and refer the details to the reference [9].
  3. Line 113. Why do the authors add H2O2? Does the system only work in an acidic environment? Add a comment in the text.
  4. Line 116. What type of lamp was used for the photocalysis measurements (wavelength, filters, power)? Is the power density reported by the authors comparable to that of the sun in the same wavelength range? Add a comment in the text.
  5. Line 151. How does the ratio of intensities of peaks D and G vary? Add a comment in the text.
  6. Figure 3c. Gap values were calculated through Kulbelka-Munk. While the tangent to the curve was considered for the alpha-Fe2O3 sample, it is unclear how the slope was calculated for the composite sample. Considering also in this case the tangent to the curve, a smaller gap value would be obtained. Add a comment in the text.
  7. Comment in Figure 4b in the text, if it allows for additional information. Otherwise remove it.
  8. Line 201 (Figure 4c). How was the material recovered to repeat 5 cycles? Was a wash carried out between one cycle and another? Add a comment in the text.
  9. Figure 5. In the case of phenols, what peak was followed in the photocatalysis process? Is it real photodegradation, or is it the breakdown of the molecule into equally toxic compounds? Add a comment in the text.
  10. Line 220. What kind of charge is transferred? Are the electrons being captured by the defects in the CNTs? Do these electrons participate in the photocatalysis process (equation 2 and 4) or do they slow down the e / h recombination process in alpha-Fe2O3? Add a comment in the text.

Reviewer 2 Report

Review

The manuscript "One-step Construction of Multi-walled CNTs Loaded with Alpha-Fe2O3 Nanoparticles for Efficient Photocatalytic Properties" by Jianle Xu et al. Is devoted to solving an urgent problem for photocatalysis aimed at reducing the recombination of photoinduced electron-hole pairs.

Before accepting an article for publication in the journal, it is necessary to clarify and correct some aspects.

  1. The introduction to the article does not quite accurately anticipate and emphasize the relevance and significance of the research. I recommend in the introduction, instead of links to review articles, to give a small overview of the synthesis methods:
  2. A) pure Alpha-Fe2O3 used as a photocatalyst (for example, https://doi.org/10.1039/C9RA07490B, 1166/JNN.2017.14199)

this is necessary because the title of the article emphasizes the one-step synthesis method as an advantage.

  1. B) tell what concepts exist for Alpha-Fe2O3 to limit recombination….

  1. There are many works in the literature on the synthesis of Multi-walled CNTs / Alpha-Fe2O3 nanocomposites, which are also used for photocatalysis. It is necessary to mention such works (10.20964/2019.01.09, 10.1088/1742-6596/1082/1/012048,

https://doi.org/10.2991/icmmcce-15.2015.444)

  1. The authors write that Alpha-Fe2O3 nanoparticles were synthesized according to the method described in “Li, P.W .; Yan, X.L .; He, Z.Q .; Ji, J.L .; Hu, J .; Li, G .; Lian, K .; Zhang, W. D .; Alpha-Fe2O3 concave and hollow nanocrystals: top-down etching synthesis and their comparative photocatalytic activities. Crystengcomm 2016 18, 1752-1759. " However, firstly, it is not clear by what particular methodology, and secondly, different reagents were used in the works ... This is also evidenced by the discrepancy in the sizes of nanoparticles. In this work, it is said about the size of the order of 20-30 nm, while in the previous work, the minimum diameter of nanoparticles is 146 nm. Therefore, I strongly recommend to clarify the synthesis method so as not to mislead the readers.

  1. It is necessary to bring the SEM image of pure Alpha-Fe2O3.

  1. Why is H2O2 added? When hydrogen peroxide is added, a heterogeneous photofenton process can occur. Nothing is said about this in the work. What do the authors think about this? I recommend showing the results of the photocatalytic decomposition of Rhodamine B without the addition of hydrogen peroxide.

I also recommend conducting experiments on the lifetime of at least the most effective photocatalyst.

  1. It is necessary to indicate how the concentration of Rhodamine B changes during photolysis without a photocatalyst.

  1. How the photocatalyst was separated from the Rhodamine B.

  1. In Figure 2 b, c, the wavelength scale and hv must be specified within the same limits. If 1.8 - 2.6 eV then respectively 730-460 nm or vice versa. It is not clear where the 2.2 eV peak (563 nm and more) for pure Alpha-Fe2O3 came from and what it is associated with, since the absorption spectrum ends at 560 nm. How do the authors explain this?

The peak at 457 nm in Figure 2b is incorrect

  1. The text omits subscripts and superscripts everywhere, please be more careful (Lines 183, 161, 216, 221, etc.)

  1. In many places the designation CNTs / α-α-Fe2O3-Sx is given. Apparently by mistake, please fix it.

  1. Judging by the given possible mechanism of the occurring reactions, does hydrogen peroxide take no part in the process? Why was it added then?

In equation 2, the index is mixed up for oxygen.

Round 2

Reviewer 2 Report

I agree with the main corrections of the authors. The article can be recommended for publication in the journal Materials.

Still, I would recommend, if you are talking about a photostimulated heterogeneous fenton-like catalysis, at least in the schematic equations of the reaction to reflect the possible processes.